# Solvent-driven fractional crystallization for atom-efficient separation of metal salts from permanent magnet leachates

Caleb Stetson [1], Denis Prodius[2], Hyeonseok Lee [1], Christopher Orme [1], Byron White[1], Harry Rollins [1], Daniel Ginosar[1], Ikenna C. Nlebedim [2] & Aaron D. Wilson [1✉]

This work reports a dimethyl ether-driven fractional crystallization process for separating rare earth elements and transition metals. The process has been successfully applied in the treatment of rare earth element-bearing permanent magnet leachates as an atom-efficient, reagent-free separation method. Using ~5 bar pressure, the solvent was dissolved into the aqueous system to displace the contained metal salts as solid precipitates. Treatments at distinct temperatures ranging from 20–31 °C enable crystallization of either lanthanide-rich or transition metal-rich products, with single-stage solute recovery of up to 95.9% and a separation factor as high as 704. Separation factors increase with solution purity, suggesting feasibility for eco-friendly solution treatments in series and parallel to purify aqueous material streams. Staged treatments are demonstrated as capable of further improving the separation factor and purity of crystallized products. Upon completion of a crystallization, the solvent can be recovered with high efficiency at ambient pressure. This separation process involves low energy and reagent requirements and does not contribute to waste generation.

[1] Critical Materials Institute, Idaho National Laboratory, 1955 N Fremont Ave, Idaho Falls, ID 83415, USA. [2] Critical Materials Institute, Ames Laboratory, US Department of Energy, Ames, IA 50011-3020, USA. ✉email: Aaron.Wilson@inl.gov

The late twentieth century saw the development of new classes of hard magnetic materials, specifically, permanent magnets containing rare earth elements (REEs, comprised of the lanthanide (Ln) series, scandium, and yttrium)[1]. REE-based permanent magnets are indispensable in many modern technologies, including electric vehicle powertrains, wind turbines, and electronic devices[2]. Beyond permanent magnets, REEs are also critical in the production of catalysts, metal alloys, polishing media, batteries, ceramics, phosphors, and glasses[3,4]. Common REE-based permanent magnets include neodymium-iron-boron (Nd-Fe-B)[5] and samarium-cobalt (Sm-Co) types[6], with usage determined by the magnetic performance and operating conditions for a particular application[1,2]. In high-temperature or corrosive environments, Sm-Co magnets are preferable based on their high coercivity, corrosion resistance, and thermal stability[6,7]. Sm-Co magnets are rich in cobalt, a critical metal with broad usage in lithium-ion batteries, metal alloys, and catalysts[8].

An electrified infrastructure and circular economy require efficient reuse of supply-limited materials such as REEs[9–12]. Extraction of REEs from secondary feedstocks like magnet scrap begins with leaching (selective, or complete)[13–17]. Once constituents of the magnet are brought into solution by leach processes, transition metal and lanthanide elements must be separated. Historically, REE separations were carried out through fractional crystallization (FC), whereby evaporation or changes in solution temperature drive precipitation of individual REE double salts from the mixed REE salt solution[18,19]. Modern lanthanide separations are accomplished via reagent-driven precipitations[20,21], solvent extraction (SX)[22,23], ion exchange (IX)[24], and electrochemical methods[25–28]. Research efforts are also underway to explore ionic liquids[29–31], biological chelating agents[32], and other methods[33] to improve REE separations. Constraints of REE purification processes, whether applied to hard rock minerals[34–36] or recycled scrap[13,37,38], generally include consumption of energy and strong acid/base reagents, generation of large volumes of wastewater, and use of complex multistep processes. The poor atom economy of these processes is directly connected to increased waste production and associated operational increases in costs, environmental risk, and potential for occupational exposure to hazards. Atom economy/efficiency is the number of atoms in a product relative to the starting material and reagents; higher efficiencies are associated with less waste and energy input[39,40]. Hydrometallurgical separations would benefit from low energy, atom-efficient, and environmentally benign processes.

FC, also referred to as antisolvent crystallization, is a separation process whereby two or more solutes are recovered from a multicomponent solution[41,42]. FC is atom-efficient and can be driven through solvent removal (evaporation), temperature change, chemical reactions, pH adjustment, or use of an additional solvent (often termed antisolvent)[41–54]. While evaporatively driven precipitation processes suffer from low energy efficiency and water consumption[55], thermally driven precipitation (e.g., solution cooling) has limited recoveries determined by temperature-dependent solubilities of solutes and requires nearly saturated solutions[56]. Chemical reactions and pH adjustment inherently consume chemicals and produce waste. Contemporary research in solvent-driven FC for mineral processing has employed alcohols and acetone to induce FC in mixed salt solutions to recover scandium[57,58] and REE salts[59,60]; however, as in historical treatments of brines, these processes may be limited by post FC requirements for separation of water and solvent[48].

Prior work determined that solvent-driven FC is a molar displacement process[61] rather than a dielectrically driven process. Regardless of the organic solvent employed, the same molar fraction of NaCl was precipitated from a saturated solution, indicating that selecting for a low molecular mass solvent is preferable to selecting for a low dielectric solvent[61]. Moreover, salts with high molar solubilities[62–64] require more solvent to be displaced in FC than salts with an equivalent mass but lower molar solubilities[61]. High molar solubility salts also induce liquid-liquid equilibrium (LLE) separation of the organic solvent, limiting salt crystallization. For example, solvent-driven FC applied to saturated NaCl solutions results in only ~12% of the NaCl being crystallized. Salts with low molar solubilities do not induce LLE separation of the organic solvent, as demonstrated in the precipitation of sparingly soluble salts (98 wt% $CaSO_4$ crystallized from a saturated solution)[61]. Many transition metal and lanthanide salts have relatively high mass fraction solubilities (>25 wt%) but rather low molar solubilities (<2.5 molal), making them an interesting target for solvent-driven FC with the potential for high recoveries. Based on these findings, we have deployed solvent-driven FC to separate transition metal and lanthanide salts using dimethyl ether (DME), a low molecular mass non-coordinating solvent that can be easily recovered[65–68].

DME is a hygroscopic gas that is partially miscible with water when condensed[69]. Historically, DME has attracted interest as both a clean-burning alternative fuel[70] and as a refrigerant capable of high performance at moderate pressures[69]. Herein, DME-driven FC is demonstrated in the treatment of REE-containing permanent magnet leachates to selectively precipitate either transition metal-rich or lanthanide-rich crystalline solid products. The leachate was produced in an acid-free magnet dissolution process[71,72] by dissolving the permanent magnets in a copper (II) sulfate solution. These separations are facilitated by the differing solubility responses of transition metal and lanthanide sulfates to temperature change; in a temperature range from 20–50 °C, transition metal sulfate solubilities in $H_2O$ increase while lanthanide sulfate solubilities decrease[73–75]. These temperature and pressure adjustments to the mixed salt aqueous solution, including the temperature and pressure swing for DME recycling, represent a trivial energy requirement when compared to thermal-[76,77] or solar-driven[78] evaporative precipitation processes.

In this work, we present an energy- and reagent-efficient solvent-driven FC process that avoids the generation of waste byproducts for the separation of lanthanides and transition metal salts from permanent magnet leachates. This separation is demonstrated for two example feedstocks: a moderately complex feedstock containing Sm, Co, and Fe and another more complex feedstock comprised of Nd, Pr, Dy, Sm, Fe, and Co. Since DME is non-toxic and readily leaves the solution as a gas at ambient temperatures and pressures for later reuse, no reagents are consumed, and no toxicity is created in the separation process.

## Results and discussion

**Experimental approach.** Gaseous DME is employed as a saturation agent to induce crystallization of transition metal or lanthanide sulfates from mixed metal salt magnet leachates. DME is not thought to interact directly with other solutes; instead, DME reduces the quantity of the free water (i.e., water that is not bound within a solvation environment) to fulfill its own hydration requirements[62,63]. A reduction in free water, reduced water activity, and/or liquid phase microstructuring[79] induce salt precipitation. This mechanism suggests that the solid salt product compositions would be similar (if not identical) to those produced in energy-intensive evaporative precipitation processes conducted at equivalent temperatures. Increasing reaction chamber head pressure increases the amount of DME dissolved into solution[80,81], resulting in crystallization of a metal salt. The experimental apparatus is depicted conceptually in Fig. 1a; a jacketed glass vessel sealed at both ends and connected with

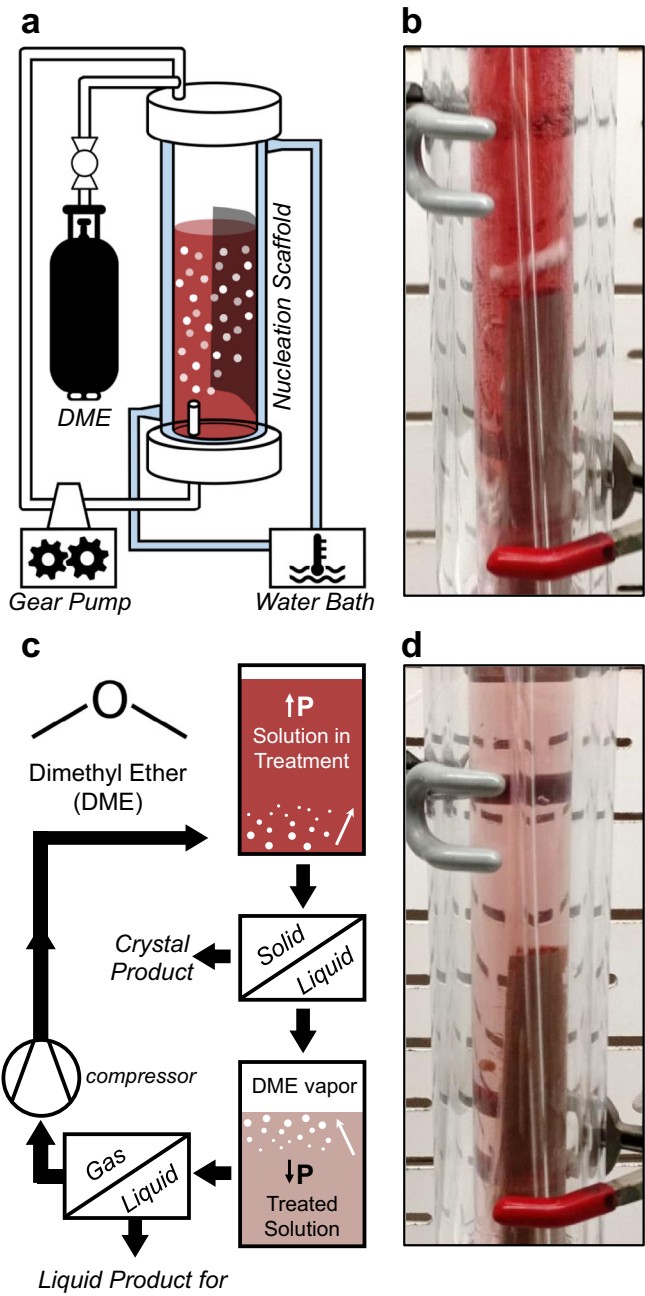

**Fig. 1 DME-FC apparatus and process schematic. a** Schematic depicting the experimental apparatus whereby DME gas is sparged into an aqueous solution at elevated pressure, permitting dissolution of DME into the liquid. Reaction temperature is controlled via a water bath, recirculation is carried out through a gear pump, and crystallization of metal salts occurs on the nucleation scaffold. **b** Photograph of the experimental apparatus during treatment of the Sm-Co leachate. **c** Process schematic depicting the DME-FC solid-liquid separation followed by a gas-liquid separation to recover and reuse DME with high efficiency. **d** Photograph of the experimental apparatus after FC of $CoSO_4$ from the leachate, showing the visible change in $CoSO_4$ concentration concurrent with crystal growth on the nucleation scaffold.

scaffold in contact with the liquid phase, depleting metal ion salt(s) from the aqueous solution. Within the reactor, we employ stainless-steel mesh as a nucleation scaffold to increase the nucleation density and facilitate the recovery of the crystallization products[82]. Once the liquid phase is evacuated from the reaction column, precipitates are captured on, and subsequently recovered from, the nucleation scaffold. By reducing the pressure exerted on the treated solution, gaseous DME can be recovered efficiently for reuse (see process flowsheet, Fig. 1c). The treated solution (Fig. 1d) remains unchanged in its chemical character (i.e., pH, salts contained in solution), apart from the removal of pre-cipitated salts, and is thus suitable for reuse in leaching or other hydrometallurgical processes. Experiments were conducted at ~62 psig, pursuant to the internal vapor pressure of the DME tank at ambient temperatures.

DME-FC was studied experimentally in conjunction with two separate REE-rich leachates: one leachate was produced from Sm-Co magnet swarf, whereas a more complex leachate was generated from a real-world mixed magnet recycling feed, containing both Nd-Fe-B and Sm-Co magnet swarf. The sensitivity of the solutes to temperature shifts[73–75] (Fig. 2a) facilitates the separation of transition metal and lanthanide sulfates from the mixed metal salt solutions. Divergent aqueous solubility vs. temperature trends exist for the metal sulfates contained in the two leachates: from 10 to 50 °C, lanthanide sulfate solubilities decrease with temperature while transition metal sulfate solubilities increase (Fig. 2a)[73–75]. Initial metal concentrations in the Sm-Co leachate (Fig. 2b) and Nd-Fe-B mixed magnet leachate (Fig. 2c) were measured by inductively coupled plasma optical emission spectroscopy (ICP-OES). Experimental observations regarding DME-FC rate and related crystal size suggest that in this system, crystallization rate and crystal size vary with temperature; at lower temperatures (e.g., 20 °C), the crystallization rate is enhanced by the increased concentration of DME in water[80,81] as determined by the headspace pressure of the DME tank, ~62 psig[81]. Higher solution viscosity at lower temperatures[83] may also lead to greater turbulence in gas bubble flow and more rapid dissolution of DME into the aqueous phase[84].

**DME-driven fractional crystallizations under controlled temperatures.** DME-FC was conducted on the leachates at two separate temperatures. In FC treatments applied to both leachates (initial concentrations given in Fig. 2b, c) at 31 °C, the increased solubility of transition metal sulfates ($FeSO_4$ and $CoSO_4$) maintains Fe and Co in solution and motivates crystallization of Ln-rich products (Fig. 3a, b). Conversely, treatment of the Sm-Co leachate at a lower temperature of 20 °C leverages the increased solubility of Sm in solution to preferentially crystallize the higher value Co fraction on the stainless-steel scaffold (Fig. 3a). This ability to combine solvent-driven and temperature-driven FC is advantageous over evaporatively driven precipitations, where temperature control is more complex. However, it may also be more challenging to control concentration gradients in evapora-tively driven processes, as during the evaporation of water, the solute must be redistributed at a diffusion rate that matches or exceeds rates of nucleation processes to ensure uniform behavior[85]. In contrast, DME is distributed through a salt solu-tion more rapidly and uniformly than a precipitation process in the studied systems, as demonstrated by the increase in the aqueous solution volume long before turbidity or macroscopic crystals are observed.

Separation efficacy for DME-FC was quantified with a separation factor, $\alpha$, a common method to evaluate SX and IX processes[86,87]. The separation factor is the ratio of distribution

fittings, valves, and tubing permits DME sparging of the leachate within the inner chamber while reaction temperature is con-trolled in the outer chamber via a water bath. In the photograph shown in Fig. 1b, the solution becomes saturated with DME, inducing crystallization that occurs primarily on the nucleation

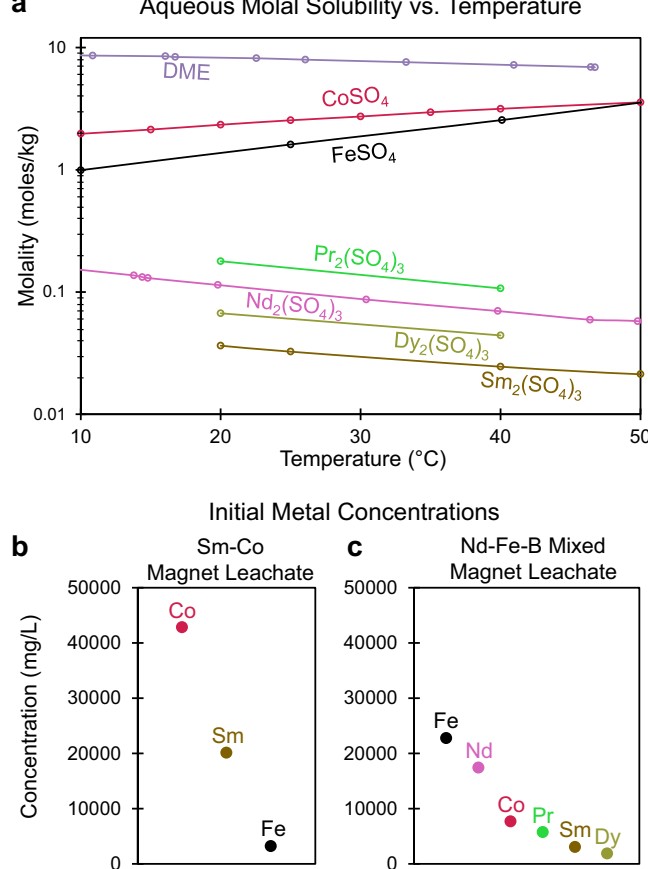

**Fig. 2 Solubility vs. temperature trends for dissolved species and initial leachate metal concentrations. a** Molal solubility limits vs. temperature for DME[81] (at sufficient pressure to condense a liquid DME phase) and metal sulfates contained in the two studied permanent magnet leachates in the temperature range from 10–50 °C[73-75]. ICP-OES measurement of metal concentrations in the **b** Sm-Co magnet leachate and **c** Nd-Fe-B mixed magnet leachate. Source data are provided as a Source Data file.

coefficients, $K_d'$, as determined in this work by the ratio of metal in the solid product relative to that in the original aqueous phase, Eq. (1):

$$\alpha_{Co/Sm} = \frac{K_{dCo}'}{K_{dSm}'} = \frac{\frac{\text{Co Mass \% in Solid Product}}{\text{Co Mass \% in Initial Aqueous}}}{\frac{\text{Sm Mass \% in Solid Product}}{\text{Sm Mass \% in Initial Aqueous}}} \quad (1)$$

Separation factors for transition metal—lanthanide separations range from 48–730 (see Table 1), with the highest selectivity achieved in the crystallization of the Sm-rich product from the Sm-Co magnet leachate.

X-ray diffraction (XRD) patterns of known references[88,89] were compared to experimental XRD datasets obtained for Co-rich solids and Sm-rich solids, showing agreement with CoSO$_4$·6H$_2$O[88] and Sm$_2$(SO$_4$)$_3$·8H$_2$O[89] (Supplementary Fig. 1). These results indicate that the solid products are recovered as sulfates, corresponding to their solubilized form in the aqueous leachate.

To quantify the recovery efficacy of DME-FC in the treatment of the two leachates, recovery fractions were calculated based on the concentration of metals in the original and treated solutions. ICP-OES results for treated Sm-Co leach solutions show 95.9% Co recovery in the 20 °C treatment and 62.5% Sm recovery in the 31 °C treatment (see Supplementary Fig. 2). Results for the Nd-Fe-B mixed magnet leach solution

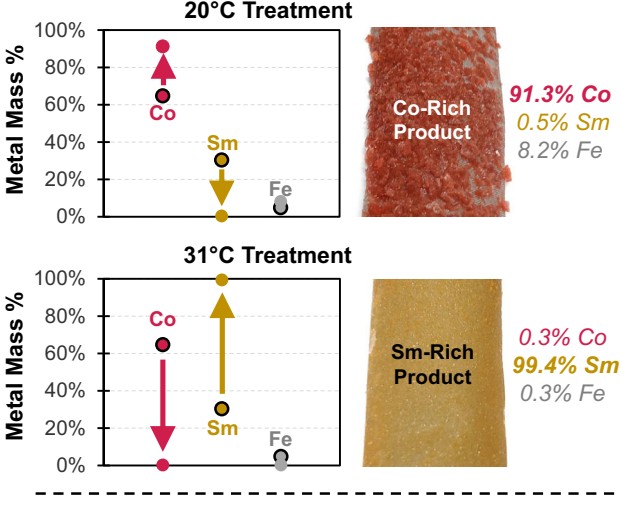

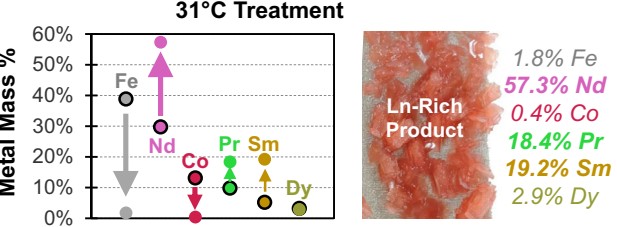

**Fig. 3 Compositional data for solid products of DME-FC. a** Solid products obtained from DME-FC of the Sm-Co magnet leachate at 20 and 31 °C. **b** Solid product obtained from DME-FC of the Nd-Fe-B mixed magnet leachate at 31 °C. ICP-OES acquired mass percent compositions are plotted at left (arrows indicate the shift from the original leachate composition to solid product composition). Associated separation factors are listed in Table 1. Photographs of the solid products as precipitated on nucleation scaffolds are shown at right. Source data are provided as a Source Data file.

treated at 31 °C show 76.1% Ln recovery. These high metal recoveries are consistent with a process determined by molar solubilities of the salt rather than mass-based solubilities[61]. Experiments were ended once extensive solid product had formed; as such, it is unlikely that these recovery fractions represent a thermodynamic endpoint. Prolonged experiments were avoided; once a fraction of the crystallizing salt has been depleted from leachate, the solution composition has changed such that a different metal salt composition is preferred in crystallization. Under such circumstances, the subsequent FC product is no longer representative of treatment of the initial leach solution, and separate metal salts may crystallize on distinct surfaces (an example optical microscope image is shown in Supplementary Fig. 3). The simultaneous growth of multiple crystal types in spatially distinct locations indicates that the product crystal structure is also a factor in the process selectivity. Further investigation is required to determine the tradeoffs between recovery fraction and product purity and to model the separation mechanism. It is important to note that changes in solution composition do not fully balance with the compositions of the sampled crystals; this may occur due to changes in the solution temperature and pressure during the evacuation of the reaction chamber, precipitation losses within sampling hardware, or simultaneous growth of differing

crystals. Complete compositional data for original and treated solutions is tabulated within Supplementary Table 1.

In the Co-rich solid product, Sm is largely excluded; however, the Co:Fe ratio is similar to that of the original leachate (11.13 vs. 13.20). This is likely a reflection of the similar crystallographic structures of $CoSO_4$ and $FeSO_4$[90,91], which permits their co-crystallization within the same crystal lattice[92]. This is supported by minimal Co and Fe entrainment in the trivalent $Sm_2(SO_4)_3$

**Table 1 Separation factors for DME-FC treatments of Sm-Co magnet leachate and Nd-Fe-B mixed magnet leachate for products depicted in Fig. 3.**

| Leachate | Temperature | Product | Separation | $\alpha\ (\ell_0 \to s_1)$ |
|---|---|---|---|---|
| Sm-Co | 20 °C | Co-rich | $\alpha_{Co/Sm}$ | 95.3 |
| | | | $\alpha_{Fe+Co/Sm}$ | 86.9 |
| | | | $\alpha_{Fe/Co}$ | 1.19 |
| Sm-Co | 31 °C | Sm-rich | $\alpha_{Sm/Fe+Co}$ | 379 |
| | | | $\alpha_{Sm/Co}$ | 704 |
| | | | $\alpha_{Fe/Co}$ | 13.2 |
| Nd-Fe-B | 31 °C | Ln-rich | $\alpha_{Ln/Fe+Co}$ | 48.5 |
| | | | $\alpha_{Fe/Co}$ | 1.46 |
| | | | $\alpha_{Nd/\Sigma}$ | 3.17 |
| | | | $\alpha_{Pr/\Sigma}$ | 2.07 |
| | | | $\alpha_{Sm/\Sigma}$ | 4.34 |
| | | | $\alpha_{Dy/\Sigma}$ | 0.89 |

Initial leachate compositions are found in Fig. 2b, c and Supplementary Table 1. Product compositions are found in Fig. 3a, b and Supplementary Table 1.

crystalline product, which is not amenable to $Co/FeSO_4$ crystallization within its lattice. Compositional data indicate that initial solution compositions affect separation factors of crystallizations. Moreover, solute solubilities are more deterministic of crystallized product than the solute concentration or its nearness to saturation. Process temperature also plays an important role; for example, at 20 °C, transition metal sulfates are produced, while at 31 °C, lanthanide sulfates are produced. The results also highlight the importance of product crystal structure, as it is possible to crystallize more than one metal salt within the same solid product (e.g., $Fe_{0.x}Co_{0.y}SO_4$).

**Sequential fractional crystallizations in stages and passes.** Metal-rich solutions can be treated via two distinct sequential methods, in passes to precipitate distinct fractions from a single solution, and also in stages by dissolving a solid product and treating the resulting solution (see Fig. 4a). In DME-FC passes, a solution is exposed to a set of conditions defined by temperature, DME pressure/concentration, and treatment duration/salt recovery fraction to produce an initial precipitate product. This solution can then be treated under differing conditions to produce additional solid product(s). In this way, a lanthanide-rich product can be initially precipitated at higher temperatures and removed from the solution, followed by a subsequent crystallization at lower temperatures to isolate a transition metal-rich product. This protocol is demonstrated in the treatment of the Sm-Co magnet leachate via ICP-OES measurement of metal concentrations in treated solutions in Supplementary Fig. 4.

In DME-FC stages, initial solid products are dissolved in water and resulting solutions are treated in a subsequent crystallization,

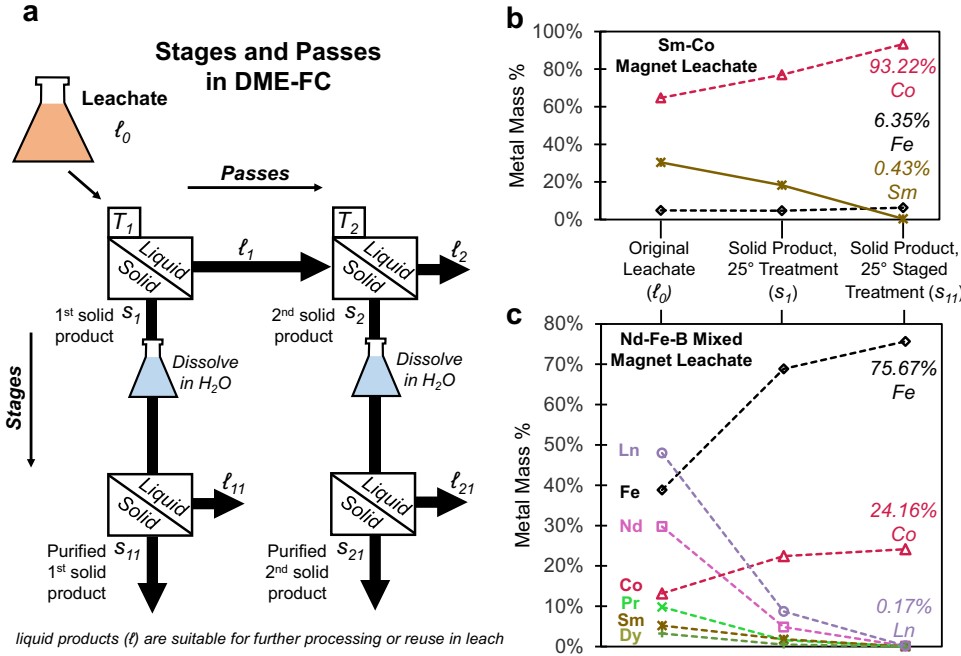

**Fig. 4 Sequential treatments in stages and passes in DME-FC. a** Scheme depicting stages and passes in DME-FC. DME-FC stages involve treatment of the initial leachate ($\ell_0$), dissolution of the the solid product (e.g., $s_1$) in water, followed by treatment of the resulting solution to yield a higher-purity solid (e.g., $s_{11}$). DME-FC passes are successive treatments of the liquid stream to recover chemically distinct solid products ($s_1$, $s_2$) from the same aqueous solution. Temperature control can enhance separations in passes (e.g., initial solid fraction $s_1$ recovered at temperature $T_1$, followed by treatment at a different temperature ($T_2$) to recover a separate solid fraction ($s_2$)), and can also be used to enhance purification via DME-FC stages. As the chemical character of the solution remains unchanged, albeit with reduced salt concentration, liquid products are suitable for further hydrometallurgical processes upstream or downstream of the separation. ICP-OES metal compositions depicting the purification from original solution ($\ell_0$), first solid product ($s_1$), and solid product produced through staging ($s_{11}$) with treatments at 25 °C for **b** Sm-Co magnet leachate and **c** Nd-Fe-B mixed magnet leachate. Sum of the lanthanide elements (Nd, Pr, Sm, and Dy) is given as Ln. Separation factors for both **b** and **c** are listed in Table 2. Source data are provided as a Source Data file.

**Table 2 Separation factors for staged treatments of the Sm-Co magnet leachate and the Nd-Fe-B mixed magnet leachate at 25 °C as depicted in Fig. 4.**

| Leachate | Product | Separation | $\alpha_1$ | $\alpha_2$ | $\alpha_{total}$ |
|---|---|---|---|---|---|
| | | | $\ell_0 \to s_1$ | $s_1 \to s_{11}$ | $\ell_0 \to s_{11}$ |
| Sm-Co | Co-rich | $\alpha_{Fe/Co}$ | 0.82 | 1.11 | 0.91 |
| Sm-Co | Co-rich | $\alpha_{Fe+Co/Sm}$ | 1.96 | 52.3 | 102 |
| Nd-Fe-B | Fe + Co-rich | $\alpha_{Fe+Co/Ln}$ | 9.69 | 54.5 | 528 |
| Nd-Fe-B | Fe + Co-rich | $\alpha_{Fe/Co}$ | 1.04 | 1.02 | 1.06 |

resulting in two-stage solid products of increased purity. To investigate this process, an intermediate temperature of 25 °C was selected to promote uniform crystal growth. DME-FC experiments conducted at 25 °C yielded solid products that were dissolved in water to saturation, then treated again under the same conditions (Fig. 4a). The sequential product obtained from the Sm-Co leachate was Co-rich, with higher purity than the treatment carried out at 20 °C (Fig. 4b). Similarly, sequential treatment of the Nd-Fe-B mixed magnet leachate at 25 °C produced a Fe- and Co-rich solid product with minimal Ln entrainment (Fig. 4c). In hydrometallurgical extractions of REEs from primary feedstocks, the leachate contains large amounts of Fe that must be precipitated through pH neutralization[93] or carried over with the lanthanides during SX[18]. DME-FC may provide a means to selectively precipitate Fe from REE-bearing solutions without chemical consumption before SX, reducing the extractant required to load REE onto organic.

Separation factors calculated for two-stage DME-FC are given in Table 2. The total separation factors for separating the transition metals from lanthanides were measured to be 528 in the Nd-Fe-B mixed magnet leachate and 102 for the Sm-Co magnet leachate. Separation factors for the two-stage process were measured as 9.69 and 54.5 in the Nd-Fe-B mixed magnet leachate, and 1.96 and 52.3 in the Sm-Co magnet leachate. This progressive increase in separation factor highlights the impact of the initial solution composition on separation efficacy: in DME-FC, selectivity increases with the concentration of the major component. In many separations, the selectivity of the process declines with increasing concentration[94]. In membrane processes, the rejection of the membrane declines as concentration increases[95], increasing the concentration of minor components in the liquor. When separation factors are relatively independent of concentration due to chemical interactions, as is the case with SX and IX, increasing purity becomes more difficult as the minor component declines. In the case of DME-FC, the separation factor is enhanced as the minor component declines. This ability of crystallization processes to access high purity is commonly used in industrial processes such as float zone refining of single crystal silicon[96].

Separation factors for transition metal (Co/Fe) separations are relatively low for both systems. Many factors influence these results, including the presence of the lanthanide ions and relative concentrations of Fe and Co in the initial solutions. While Fe and Co concentrations are within an order of magnitude in the studied solutions, molal solubility of $CoSO_4$ is twice that of $FeSO_4$ at 25 °C. Given that Fe tends to be incorporated in transition metal-rich solid products (even where it appears as a minor component, as in the Sm-Co magnet leachate), small solubility differences can have a significant impact on DME-FC separations.

In summary, DME-driven FC was demonstrated in the separation of REE and transition metal salts from industrially generated magnet wastes. DME-FC was applied to two separate leachates, one comprising only Sm, Co, and Fe, and a more complex leachate containing Nd, Pr, Dy, Sm, Fe, and Co. Depending on the temperature (20–31 °C), the process can be tailored to selectively yield either transition metal-rich or lanthanide-rich solid products. High recoveries are observed for separations obtained in DME-FC (62.5–95.9% recovery), indicating that high-yield sequential processing can be achieved with a limited number of steps. Staged treatment of the leachate, involving dissolution of the solid products and treatment of the resulting solutions at 25 °C, produced high purity transition metal salt products (>99.5% transition metal). The selectivity of the process increases with the concentration of the major component, suggesting that DME-FC may be an effective processing route to generate high purity products.

DME-driven FC offers non-toxic separation of valuable elements from a mixed salt solution, avoiding requirements of stoichiometric chemical consumption. The selectivity in solid products avoids the energy costs of distillation and the introduction of additional ions to the working fluid (salt metathesis). This process presents opportunities for significant reductions in downstream environmental effects associated with state-of-the-art hydrometallurgical purifications. DME-driven FC is a versatile separation that can be integrated with existing separations such as SX to reduce reagent usage, waste generation, and energy consumption.

## Methods

**Leachate preparation.** Leach solutions were obtained by oxidative dissolution of Sm-Co magnet grinding swarf (1.0 kg) and a mixture of Sm-Co and Nd-Fe-B grinding swarfs supplied by a U.S. magnet processing plant (1.0 kg). 3.46 kg of copper(II) sulfate ($CuSO_4$, 12 wt% solution) was used for oxidative dissolution of Sm-Co magnet grinding swarfs. For the mixture of Sm-Co and Nd-Fe-B grinding swarfs, 2.03 kg of copper(II) sulfate ($CuSO_4$, 10 wt% solution) was used. The dissolution reaction was initiated at ambient temperature and proceeded exothermically with stirring applied for 5 h at 300 rpm. The procedure for oxidative dissolution of magnets to produce REE-rich leachates is also described in a prior publication[71].

**Experimental apparatus.** The glass reaction vessel used in treatments is a dual chambered tube; the inner tube contains the reaction while the volume in between the two tubes accommodates flow from a water bath, delivering temperature control for the reaction. The glass vessel length is ~40 cm; the inner tube measures 31.7 mm in outer diameter with 4 mm wall thickness. The outer tube measures 50 mm in outer diameter with 5 mm wall thickness. Both ends of the glass reaction vessel are threaded to accommodate Teflon endcaps, which are integrated to a recirculation system via 316 stainless steel Swagelok fittings and 1/8" Teflon tubing. Recirculation and sample introduction are accomplished through use of a gear pump (Cole Parmer 115 V 60 Hz console drive, EW 35215-30, fitted with a Cole Parmer Micro pump head, EW-07001-40). Reaction temperature is moderated with a water bath (Cole Parmer polystat standard 3–6 l capacity Heating/Cooling bath, PN: EW12122-02). Gaseous DME is introduced to the system via a standard gas cylinder (Matheson, 99.5% purity, size 1A). Stainless-steel wire mesh screen (316, 400 mesh, 0.0012" x 48" roll, purchased from wirescreen.org, 400x400T0012W48T) is cut to size and implanted in the reaction chamber to serve as a nucleation scaffold.

**Experimental conditions.** After assembly, the apparatus was pressurized and leak tested with gaseous DME, then purged (five cycles) with DME to remove partial pressures of atmospheric gas. The aqueous sample (generally 150–200 ml) was then introduced to the chamber via the gear pump. The reaction chamber was pressurized, with DME gas recirculated from the headspace through the aqueous solution. DME headspace pressure is maintained via delivery from the DME gas canister. Temperature was varied in a range from 20 °C to 31 °C by control of the water bath. Pressure was held at 62 psig, via delivery of gas pressure from the DME canister. A description of a complete DME-FC treatment is given in Supplementary Note 1: Example Experiment within the Supplementary Information.

**Inductively coupled plasma optical emission spectroscopy (ICP-OES).** Aqueous and acid-digested crystal samples were analyzed for the elements Co, Fe, Sm, Nd, Pr, and Dy with ICP-OES. The instrument used was an iCAP Series 6000 (Thermo Scientific), calibrated from commercial stock solutions (1000 ppm, five-point calibration, 0, 1, 2, 5, 10 and 20 ppm calibration range). A continuing calibration verification standard (CCV) and a continuing calibration blank are analyzed throughout the analysis to monitor instrument performance. For quality

control and assurance purposes, commercial stock solutions for the calibration standards and CCV standards are obtained from separate vendors. A laboratory control standard (LCS, or sample spike) was analyzed to ensure analytical accuracy and precision. An internal standard for the ICP-OES analysis (scandium, 4 ppm) was conducted to correct for any matrix artefacts. Concentrated hydrochloric acid (HCl) was used to produce a 2% HCl acid matrix.

Samples were prepared first by filtration through a 0.45 μm filter to remove suspended solids that may otherwise induce sample introduction problems. Solid samples were digested according to the following protocol: (1) weigh out 0.2–0.3 g of sample into a tarred Teflon 25 ml beaker, (2) rinse the side of the beaker with Nanopure $H_2O$ to collect all the sample in the bottom of the vessel (~0.5 ml), (3) add 2 ml HCl to the beaker with the sample, (4) place sample on hotplate, cover with a Teflon watchglass and heat sample, (5) once the sample has dissolved, remove the beaker from the hotplate and cool the sample, (6) analytically transfer the sample to a 25 ml poly volumetric flask and dilute to volume with Nanopure $H_2O$, (7) transfer the sample to a 50 ml poly tube which has been labeled with the sample ID.

**Optical microscopy**. Optical microscope images were captured with a camera and an Olympus Optical SZH-ILLD Lab Zoom stereo microscope. A Cole-Parmer standard fiber optic illuminator (41720 series) was utilized with the microscope.

**X-ray diffraction (XRD)**. XRD patterns were obtained for crystals recovered from nucleation scaffolds after grinding in a mortar and pestle. A Bruker D8 Advance was used in analysis; a Cu anode generated Cu K-α x-rays and 2θ was measured in a range of 5° to 90°. XRD patterns were background subtracted using Bruker Instrument Software. Match! software was applied in Reitveld refinement based on known references[88,89].

## Data availability

All data presented in this study are included in this published article, its Supplementary Information, and Supplementary Source Data file. Data presented in this study include previously reported results[73–75,81,88,89]. Source data are provided with this paper.

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

## Acknowledgements

This work was supported by the Critical Materials Institute, an Energy Innovation Hub of the U.S. Department of Energy, Office of Energy Efficiency and Renewable Energy, Advanced Manufacturing Office. The work at Idaho National Laboratory was conducted under contract DE-AC07-05ID14517. Work was performed, in part, at Ames Laboratory, operated for the U.S. Department of Energy by the Iowa State University of Science and Technology under Contract No. DE-AC02-07CH11358.

## Author contributions

C.S. performed experiments and wrote the manuscript. D.P. and I.C.N. prepared source materials and edited the manuscript. H.L. provided laboratory assistance. C.O. oversaw laboratory work and developed experimental apparatus. B.W. conducted ICP-OES. H.R. conducted XRD. D.G. provided research funding and advice. A.D.W. provided research funding, edited the manuscript, and directed research.

## Competing interests

C.S., H.L., C.O., D.G., and A.D.W. are named inventors on patent US 11,261,111 and patent application PCT/US2022/072792 associated with the described DME-FC processes the details of which are listed below. The remaining authors declare no competing interests. US 11,261,111, "Methods and Systems for Treating a Liquid" Assignee: Battelle Energy Alliance. Inventors: A.D.W., Daniel S. Wendt, C.O., Birendra Adhikari, and D.G. This patent covers the general application of dimethyl ether for fractional crystallization of solutes (DME-FC) as discussed in this publication. PCT/US2022/072792 "Selective Precipitation of Metal Ion Salts from Saturated Aqueous Solutions" Applicant: Battelle Energy Alliance. Inventors: C.S., H.L., C.O., D.G., and A.D.W. This application was submitted to World Intellectual Property Organization via Patent Cooperation Treaty on June 7, 2022. Among other things this application covers the application of sequential steps and scaffolds to the fractionation of solutes with dimethyl ether as discussed in this publication.
