## [Peer Review File · Nature Communications]

Solvent-driven fractional crystallization for atom-efficient separation of metal salts from permanent magnet leachatesREVIEWER COMMENTS

Reviewer #1 (Remarks to the Author):

This work report on the use of dimethyl ether as antisolvent in precipitation of REE from a leach solution. The application is new (and a very good idea!), however the method to use a gas and adjust the pressure for it to dissolve and precipitate a compound is not novel. Please find my comments below.

- 1) I can only guess that sulphuric acid was used for leaching. The composition of the aqueous phase should be reported in the experimental section and also in the abstract.
- 2) Antisolvent precipitation of REE sulphates by using alcohols has been published by different authors the past 5 years why is none of these papers mentioned in this work?
- 4) The term "atom-efficient" is unclear to me, how is it defined?
- 5) Separation factor is mentioned in the abstract. This terminology is not commonly used for crystallization processes.
- 6) The quality of the images in the ToC figure should be improved, it is difficult to see what they illustrate.
- 7) On page 4 it is written that "Solvent-driven FP has been historically limited by the post FP requirements for separation of water and solvent." This statement is a bit odd since antisolvent crystallization is widely used in the pharmaceutical industry.
- 8) Why do the authors use the terminology "fractional precipitation" instead of only precipitation? Fractional precipitation should be used when repeated precipitation steps are used, which is not the case in this work (only one step of precipitation of REE sulphates is used).
- 9) The discussion about the dielectric constant and solute activity is very unclear. The discussion must be improved so that the reader understand what is meant.
- 10) In the results and discussion page 6, line 125 the "nucleation scaffold" is mentioned. Please describe more in detail what is meant.
- 11) On page 8 line 153- 156, it is written that "the crystallization rate and related crystal size of FP-driven crystallization vary based on temperature". Firstly, what exactly is meant by "FP-driven crystallization? Secondly, indeed the crystallization rate should depend on temperature to some extent, this is obvious however this statement becomes very general. In addition, to write that "the rate and related crystal size depend on temperature" is very vague. The crystal size and size distribution depends on several factors. How do the authors mean that the size of the crystals depend on the temperature? Do the authors mean that the crystals grow larger during a specific time if the temperature is higher because the rate is higher? The batch system will not reach equilibrium? What about the nucleation event? Surely it must have impact on the final size of the crystals in this case.
- 12) The last sentence on page 8 is not complete.
- 13) Figure 2, why is the data reported in microg/mL and not e.g. g/L which would be easier to read and grasp?
- 14) On page 10 it is written that "during the evaporation of water the solute must be redistributed at a rate that matches that of nucleation processes to ensure uniform behavior. In contrast, DME is distributed through a salt solution more rapidly than a precipitation process." Firstly, where is the proof that the DME is distributed more rapidly than the precipitation process. Secondly what does it actually mean, it is not clear at all which rates are compared here. What is meant by that the evaporation of water must be distributed at a rate that matches that of the nucleation process.

Which rates are compared here and to which purpose. These two sentences must be clarified for the reader to understand.

15) In table 1 separation factors are given. These are based on total metal concentrations and thus depend on the composition of the system, this could deserve a comment.

16) Page 13, line 214- 215 "These high metal recoveries are consistent with a process determined by molar solubilities of the salt rather than mass-based solubilities." I don't understand the difference between molar solubilities and mass-based solubilities? This sentence is very unclear, please clarify.

17) Page 13, line 218, it is written "prolonged experiments were avoided". Would it not be good to conduct long-term experiments to determine the thermodynamically most stable phase present? What about the repeatability if the results are as chaotic as it seems?

18) Page 14, line 234- 236, it is stated that "These findings indicate that selective precipitations are influenced by initial solution composition, solute solubilities, process temperatures, and product crystal structure." This is a very general and unspecific conclusion and it is not at all clear how it relates to the results, it must be rephrased.

19) Page 14, "fractional precipitation in stages" why not use "precipitation in stages" or "fractional precipitation" because fractional precipitation normally refers to precipitation in stages.

20) On page 18 finally the presence of Fe(II) is mentioned. Why so late in the manuscript and what about oxidation to Fe(III)? You will have a mixture of Fe(II) and Fe(III)?

21) I like the approach in this manuscript and the idea of the paper but I think that the text must be improved and be given more thought. I am sorry for my bad phrasing due to lack of time. I hope that you will understand my comments.

Reviewer #2 (Remarks to the Author):

This manuscript deals with the selective recovery of critical metals from permanent magnet leachates by fractional precipitation. In general, precipitation methods in hydrometallurgy shows a poor selectivity in spite of simple operations. The solvent-driven fractional precipitation method in this manuscript has a metal selectivity; therefore, this is a promising method for environment-friendly metal refinings. Although the results are clearly presented, the authors should elaborate on the points listed below.

The solvent-driven fractional precipitation method can cause a reduction of the SX operation in the metal purification process. In contrast, the mutual separation of REE after the FP operation will still be conducted by SX. Hence, an efficient separation of Nd/Sm or Nd/Dy could make this method more eco-friendly. Does this method have a potential to achieve that?

I think that the metal concentration (component) in the obtained products (Fig 3) is significantly different from that which can be expected from Fig 2a. This is due to the contribution of DME molecules? DME can distinguish between transition metals and REEs? The authors described that one mole of salt is replaced by one mole of MOS in their previous paper. Does this mole ratio depend on the metal salts? The authors should explain a role of DME during the precipitation reaction in detail.

Why do the Nd-Fe-B leachates (Table S2) contain Sm? Nd-Fe-B magnets do not generally contain Sm.

REVIEWER COMMENTS

Reviewer #1 (Remarks to the Author):

This work report on the use of dimethyl ether as antisolvent in precipitation of REE from a leach solution. The application is new (and a very good idea!), however the method to use a gas and adjust the pressure for it to dissolve and precipitate a compound is not novel. Please find my comments below.

The authors would like to thank the reviewer for the careful reading of the manuscript. We have made significant revisions to the text and figures as outlined below.

1) I can only guess that sulphuric acid was used for leaching. The composition of the aqueous phase should be reported in the experimental section and also in the abstract.

Thank you for alerting us to this imprecision in the manuscript. We have made revisions to sentence on page 5 to draw attention to the process used to produce the REE-containing leachates:

Herein, DME-driven FP is demonstrated in the treatment of REE-containing permanent magnet leachates to selectively precipitate either transition metal-rich or lanthanide-rich crystalline solid products. **The leachate was produced in an acid-free magnet dissolution process^{71,72} by dissolving the permanent magnets in a copper (II) sulfate solution.**

- 71 Prodius, D., Gandha, K., Mudring, A.-V. & Nlebedim, I. C. Sustainable Urban Mining of Critical Elements from Magnet and Electronic Wastes. *ACS Sustainable Chemistry & Engineering* **8**, 1455-1463, doi:10.1021/acssuschemeng.9b05741 (2019).
- 72 Chowdhury, N. A., Deng, S., Jin, H., Prodius, D., Sutherland, J. W. & Nlebedim, C. I. Sustainable Recycling of Rare-Earth Elements from NdFeB Magnet Swarf: Techno-Economic and Environmental Perspectives. *ACS Sustainable Chemistry & Engineering* **9**, 15915-15924, doi:10.1021/acssuschemeng.1c05965 (2021).

The Materials and Methods section contained within the SI features a more explicit description of the dissolution process to produce the leachates from magnet swarf:

Leachate Preparation

Leach solutions were obtained by oxidative dissolution of Sm-Co and Nd-Fe-B grinding swarfs (1 kg each) using copper(II) sulfate (10 wt.% solution). The dissolution reaction was initiated at ambient temperature and proceeded exothermically with stirring applied for 5 h at 300 rpm. The procedure for dissolution of magnets to produce REE-rich leachates is described in further detail in a previous publication.⁷¹

- 71 Prodius, D., Gandha, K., Mudring, A.-V. & Nlebedim, I. C. Sustainable Urban Mining of Critical Elements from Magnet and Electronic Wastes. *ACS Sustainable Chemistry & Engineering* **8**, 1455-1463, doi:10.1021/acssuschemeng.9b05741 (2019).

2) Antisolvent precipitation of REE sulphates by using alcohols has been published by different authors the past 5 years why is none of these papers mentioned in this work?

Thank you for bringing these publications to our attention. In our revisions we have modified a sentence and added four new references to introduce these recent publications on page 4:

Contemporary research in solvent-driven FC for mineral processing has employed alcohols and acetone to induce FC in mixed salt solutions to recover scandium^{57,58} and REE salts;^{59,60} however, as in historical treatments of brines, these processes may be limited by post FC requirements for separation of water and solvent.⁴⁸

- 57 Kaya, Ş. *et al.* Scandium Recovery from an Ammonium Fluoride Strip Liquor by Anti-Solvent Crystallization. *Metals* **8**, doi:10.3390/met8100767 (2018).
- 58 Peters, E. M., Svärd, M. & Forsberg, K. Impact of process parameters on product size and morphology in hydrometallurgical antisolvent crystallization. *CrystEngComm* **24**, 2851-2866, doi:10.1039/d2ce00050d (2022).
- 59 Korkmaz, K., Alemrajabi, M., Rasmuson, Å. C. & Forsberg, K. M. Separation of valuable elements from NiMH battery leach liquor via antisolvent precipitation. *Separation and Purification Technology* **234**, doi:10.1016/j.seppur.2019.115812 (2020).
- 60 Lewis, A., Chivavava, J., du Plessis, J., Smith, D. & Smith, J. in *Rare Metal Technology 2021. The Minerals, Metals & Materials Series*. (Springer, Cham., 2021).
- 48 Ireland, D. T. Solvent Precipitation of Salt. US4548614A (1985).

4) The term "atom-efficient" is unclear to me, how is it defined?

Thank you for mentioning this terminology. We have added a sentence on page 4 and inserted two additional references.

Atom economy/efficiency is the number of atoms in a product relative to the starting material and reagents; higher efficiencies are associated with less waste and energy input.^{39,40}

- 39 Trost, B. M. Atom Economy-A Challenge for Organic Synthesis : Homogeneous Catalysis Leads the Way. *Angewandte Chemie* **34**, 259-281, doi:<https://doi.org/10.1002/anie.199502591> (1995).
- 40 Trost, B. M. On Inventing Reactions for Atom Economy. *Accounts of Chemical Research* **35**, 695-705, doi:<https://doi.org/10.1021/ar010068z> (2002).

5) Separation factor is mentioned in the abstract. This terminology is not commonly used for crystallization processes.

Separation factor allows for ready comparison to other processes utilized in REE separations, including solvent extraction, ion exchange, and membrane processes.

On page 12-13, we justify the use of separation factor and define the terminology:

Separation efficacy for DME-FC was quantified with a separation factor, α , a common method to evaluate solvent extraction and ion exchange processes.^{86,87} The separation factor is the ratio of distribution coefficients, K'_d , as determined in this work by the ratio of metal in the solid product relative to that in the original aqueous phase, Equation 1:

$$\alpha_{Co/Sm} = \frac{K'_{dCo}}{K'_{dSm}} = \frac{\frac{Co \text{ Mass \% in Solid Product}}{Co \text{ Mass \% in Initial Aqueous}}}{\frac{Sm \text{ Mass \% in Solid Product}}{Sm \text{ Mass \% in Initial Aqueous}}}$$

- 86 Free, M. L. *Hydrometallurgy: Fundamentals and Applications*. (John Wiley & Sons, Inc., 2013).
- 87 Li, B. *et al.* Efficient separation and high selectivity for nickel from cobalt-solution by a novel chelating resin: Batch, column and competition investigation. *Chemical Engineering Journal* **195-196**, 31-39, doi:10.1016/j.cej.2012.04.089 (2012).

6) The quality of the images in the ToC figure should be improved, it is difficult to see what they illustrate.

Thank you for making this suggestion. We have simplified the ToC figure for clarity.

7) On page 4 it is written that "Solvent-driven FP has been historically limited by the post FP requirements for separation of water and solvent." This statement is a bit odd since antisolvent crystallization is widely used in the pharmaceutical industry.

Thank you for this comment. High-value pharmaceutical products produced with antisolvent crystallization are economical despite distillation costs; however, in the treatment of lower-value liquids such as leach solutions and brines, this downstream requirement for separation

of water and solvent can render the process more expensive than the value of the products. To clarify this distinction, we have made edits to the sentence:

Contemporary research in solvent-driven FC for mineral processing has employed alcohols and acetone to induce FC in mixed salt solutions to recover scandium^{57,58} and REE salts;^{59,60} however, as in historical treatments of brines, these processes may be limited by post FC requirements for separation of water and solvent.⁴⁸

8) Why do the authors use the terminology "fractional precipitation" instead of only precipitation? Fractional precipitation should be used when repeated precipitation steps are used, which is not the case in this work (only one step of precipitation of REE sulphates is used).

Thank you for this comment. After reflection, we have chosen to replace "fractional precipitation" with "fractional crystallization" throughout the manuscript for clarity. The use of "fractional" indicates a compositional difference between the solids produced and mother liquor. An example of this terminology exists in *fractional* distillation, where a single stage is considered to be fractional regardless of whether it is used in a cascade. An appropriate description of fractional crystallization can be found in reference 42:

"Fractional crystallization systems comprise solutes that are to be separated, and a solvent from which they are selectively crystallized. For verbal convenience the former will sometimes be called "salts," and the latter "water," letting these specific materials stand for their respective classes."

42 Fitch, B. How to Design Fractional Crystallization Processes. *Industrial and Engineering Chemistry* **62**, 6-33 (1970).

We have adapted the description of fractional crystallization on page 4 of the text to minimize confusion.

Fractional crystallization (FC), also referred to as antisolvent crystallization, is a separation process whereby two or more solutes are recovered from a multicomponent solution.^{41,42} FC is atom-efficient, and can be driven through solvent removal (evaporation), temperature change, chemical reactions, pH adjustment, or use of an additional solvent (often termed antisolvent).⁴¹⁻⁵⁴

41 Wibowo, C. & Ng, K. M. Unified Approach for Synthesizing Crystallization-Based Separation Processes. *AIChE Journal* **46**, 1400-1421 (2000).

42 Fitch, B. How to Design Fractional Crystallization Processes. *Industrial and Engineering Chemistry* **62**, 6-33 (1970).

9) The discussion about the dielectric constant and solute activity is very unclear. The discussion must be improved so that the reader understand what is meant.

Thank you for this insightful comment. In our efforts to be economic in the introduction, this was not fully explained. We have added discussion to the paragraph:

Prior work determined that solvent-driven FC is a molar displacement process⁶¹ rather than a dielectrically driven process. **Regardless of the organic solvent employed, the same molar fraction of NaCl was precipitated from a saturated solution, indicating that selecting for a low molecular mass solvent is preferable to selecting for a low dielectric solvent.⁶¹ Moreover, salts with high molar solubilities⁶²⁻⁶⁴ require more solvent to be displaced in FC than salts with an equivalent mass but lower molar solubilities.⁶¹ High molar solubility salts also induce liquid-liquid equilibrium (LLE) separation of the organic solvent, limiting salt crystallization. For example, solvent-driven FC applied to saturated NaCl solutions results in only ~12% of the NaCl being crystallized. Salts with low molar solubilities do not induce LLE separation of the organic solvent, as demonstrated in the precipitation of sparingly soluble salts (98 wt.% CaSO₄ crystallized from a saturated solution).⁶¹ Many transition metal and lanthanide salts have relatively high mass fraction solubilities (>25 wt.%) but rather low molar solubilities (<2.5 molal), making them an interesting target for solvent-driven FC with the potential for high recoveries. Based on these findings, we have deployed solvent-driven FC to separate transition metal and lanthanide salts using dimethyl ether (DME), a low molecular mass non-coordinating solvent that can be easily recovered.⁶⁵⁻⁶⁸**

10) In the results and discussion page 6, line 125 the "nucleation scaffold" is mentioned. Please describe more in detail what is meant.

Thank you for this suggestion. We have added a sentence and a reference on page 6 to better define the nucleation scaffold and its role in the reactor. In addition, text in the experimental section in the SI describes the nucleation scaffold material.

In the photograph shown in Figure 1b, the solution becomes saturated with DME, inducing crystallization that occurs primarily on the nucleation scaffold in contact with the liquid phase, depleting metal ion salt(s) from the aqueous solution. **Within the reactor, we employ stainless-steel mesh as a nucleation scaffold to increase the nucleation density and facilitate the recovery of the crystallization products.⁸²** Once the liquid phase is evacuated from the reaction column, precipitates are captured on, and subsequently recovered from, the nucleation scaffold.

82 Spinnewyn, J., Neslfidek, M. & Asinari, C. Diamond nucleation on steel substrates. *Diamond and Related Materials* **2**, 361-364 (1993).

316 stainless steel 400 mesh (0.0012" x 48" roll, purchased from wirescreen.org, 400x400T0012W48T) is cut to size and implanted in the reaction chamber to serve as a nucleation scaffold.

11) On page 8 line 153- 156, it is written that "the crystallization rate and related crystal size of FP-driven crystallization vary based on temperature". Firstly, what exactly is meant by "FP-driven crystallization"? Secondly, indeed the crystallization rate should depend on temperature to some extent, this is obvious however this statement becomes very general. In addition, to write that "the rate and related crystal size depend on temperature" is very vague. The crystal size and size distribution depends on several factors. How do the authors mean that the size of the crystals depend on the temperature? Do the authors mean that the crystals grow larger during a specific time if the temperature is higher because the rate is higher? The batch system will not reach equilibrium? What about the nucleation event? Surely it must have impact on the final size of the crystals in this case.

Thank you for identifying this text for improvement. FP-driven crystallization was a textual error and has been corrected. We intended to refer to DME-driven fractional crystallization (DME-FC).

We have modified the text to reflect that these were experimental observations during treatment of these individual systems, and do not reflect comprehensive study of these phenomena. Complex interrelationships exist between DME solubility, salt solubilities, and temperature. In future work, these topics merit systematic study. We only intend to state that with the experimental setup employed in this work, the fractional crystallizations proceed faster at lower temperatures. The text now reads:

Experimental observations regarding DME-FC rate and related crystal size suggest that in this system, crystallization rate and crystal size vary with temperature; at lower temperatures (e.g., 20°C), the **crystallization** rate is enhanced by the increased concentration of DME in water^{80,81} as determined by the headspace pressure of the DME tank, ~62 psig.⁸¹

12) The last sentence on page 8 is not complete.

Thank you for the correction. This sentence now reads:

Higher solution viscosity at lower temperatures⁸³ may also lead to greater turbulence in gas bubble flow and more rapid dissolution of DME into **the aqueous phase**.⁸⁴

13) Figure 2, why is the data reported in microg/mL and not e.g. g/L which would be easier to read and grasp?

Thank you for this suggestion. While microg/mL is a common unit for ICP analysis (standards are sold in microg/mL), we agree that a unit change would have value. We have modified Figures 2, S2, and S4 concentration axis labels to read mg/L, a more intuitive concentration for the reader that is still equivalent to ICP-measured concentrations.

14) On page 10 it is written that "during the evaporation of water the solute must be redistributed at a rate that matches that of nucleation processes to ensure uniform behavior. In contrast, DME is distributed through a salt solution more rapidly than a precipitation process." Firstly, where is the proof that the DME is distributed more rapidly than the precipitation process. Secondly what does it actually mean, it is not clear at all which rates are compared here. What is meant by that the evaporation of water must be distributed at a rate that matches that of the nucleation process. Which rates are compared here and to which purpose. These two sentences must be clarified for the reader to understand.

Thank you for this suggestion for clarification and qualification. Evaporation occurs at a liquid-vapor interface where a solution will reach super saturation if a solid-liquid equilibrium (SLE) is to be generated. The supersaturated solution must distribute to maintain uniformity. It is not certain that in all cases DME will be distributed at faster rates than the crystallization process, but in the systems being studied DME distributes rapidly into the aqueous phase based on the observed increase in aqueous phase volume long before macroscopic crystallization is observed. The text now reads:

However, it **may** also **be** more challenging to control concentration gradients in evaporatively driven processes, as during the evaporation of water the solute must be redistributed at a **diffusion** rate that matches **or exceeds** rates of nucleation processes to ensure uniform behavior.⁸⁵ In contrast, DME is distributed through a salt solution more rapidly **and uniformly** than a precipitation process **in the studied systems, as demonstrated by the increase in the aqueous solution volume long before turbidity or macroscopic crystals are observed.**

15) In table 1 separation factors are given. These are based on total metal concentrations and thus depend on the composition of the system, this could deserve a comment.

Thank you for this recommendation. Separation factors given in Table 1 are indeed a reflection of the initial solution compositions. While these compositions are depicted graphically in Figure 2b-c, they are also given numerically in Table S1. The table caption has been modified to draw attention to the location of those data:

Table 1: Separation factors for DME-FC treatments of Sm-Co magnet leachate and Nd-Fe-B mixed magnet leachate for products depicted in Figure 3. **Initial leachate compositions are found in Figure 2b-c and Table S1.**

16) Page 13, line 214- 215 "These high metal recoveries are consistent with a process determined by molar solubilities of the salt rather than mass-based solubilities." I don't understand the difference between molar solubilities and mass-based solubilities? This sentence is very unclear, please clarify.

Thank you for this request for clarification. Our current interpretation of DME-driven fractional precipitation is that it is a molar concentration-driven process rather than a mass concentration-driven process. Salts that reach high molar concentrations like NaCl can only be modestly precipitated ~12% relative to saturation prior to reaching liquid-liquid equilibrium LLE. By mass, CoSO₄ has nearly identical solubility to NaCl but due to its greater molar mass, its molar solubility is about a third of that of NaCl (and even less if activity is considered). If indeed defined by this molar phenomenon, CoSO₄ has more in common with CaSO₄, for which we obtained 98% recovery from saturation with DME-FC. Such a process would be based on differences in free energy of mixing, which depends on molar quantities. As such, we have chosen to represent aqueous solute solubilities for DME, FeSO₄, CoSO₄, Sm₂(SO₄)₃, Nd₂(SO₄)₃, Pr₂(SO₄)₃, and Dy₂(SO₄)₃ as molal solubilities in Figure 2a. Pursuant to an earlier comment from the reviewer, we have clarified the text in this passage:

Prior work determined that solvent-driven FC is a molar displacement process⁶¹ rather than a dielectrically driven process. **Regardless of the organic solvent employed, the same molar fraction of NaCl was precipitated from a saturated solution, indicating that selecting for a low molecular mass solvent is preferable to selecting for a low dielectric solvent.⁶¹ Moreover, salts with high molar solubilities⁶²⁻⁶⁴ require more solvent to be displaced in FC than salts with an equivalent mass but lower molar solubilities.⁶¹ High molar solubility salts also induce liquid-liquid equilibrium (LLE) separation of the organic solvent, limiting salt crystallization. For example, solvent-driven FC applied to saturated NaCl solutions results in only ~12% of the NaCl being crystallized. Salts with low molar solubilities do not induce LLE separation of the organic solvent, as demonstrated in the precipitation of sparingly soluble salts (98 wt.% CaSO₄ crystallized from a saturated solution).⁶¹ Many transition metal and lanthanide salts have relatively high mass fraction solubilities (>25 wt.%) but rather low molar solubilities (<2.5 molal), making them an interesting target for solvent-driven FC with the potential for high recoveries. Based on these findings, we have deployed solvent-driven FC to separate transition metal and lanthanide salts using dimethyl ether (DME), a low molecular mass non-coordinating solvent that can be easily recovered.⁶⁵⁻⁶⁸**

17) Page 13, line 218, it is written "prolonged experiments were avoided". Would it not be

good to conduct long-term experiments to determine the thermodynamically most stable phase present? What about the repeatability if the results are as chaotic as it seems?

Thank you for identifying this section for improvement. Ultimately, we are interested in conducting prolonged experiments to determine thermodynamic endpoints. However, presenting fractional precipitation results for complex mixed salt solutions in prolonged experiments may be misleading the reader; this is why we have chosen to focus this research work on defined experiments with controlled temperatures and limited DME exposure time. For example, in a preliminary experiment carried out on the Sm-Co magnet leachate, the Co fraction was largely depleted from solution, concurrent with Co-rich crystallization on the nucleation scaffold. At this stage, the leach solution had been transformed from Co-rich to Sm-rich. Prolonged exposure to DME then resulted in crystallization of Sm-rich solids in separate locations on the nucleation scaffold (see Figure S3).

In summary, we avoided investigation of prolonged experiments because after hours of treatment with DME, the solution has changed such that crystallization behavior changes dramatically, and no longer represents treatment of the original leachate. We have revised the text to clarify this issue:

Prolonged experiments were avoided; once a fraction of the crystallizing salt has been depleted from leachate, the solution composition has changed such that a different metal salt composition is preferred in crystallization. Under such circumstances, the subsequent FC product is no longer representative of treatment of the initial leach solution, and separate metal salts may crystallize on distinct surfaces (an example optical microscope image is shown in Figure S3).

We have also made revisions to the figure caption of Figure S3:

Figure S3. Optical image of two classes of precipitates on stainless-steel scaffold after prolonged DME-FC treatment at 23°C. Initially, Co-rich solids were crystallized from solution. After crystallizing the majority of the transition metal fraction (which yields an Sm-rich treated solution), several hours of prolonged exposure to DME resulted Sm-rich solids crystallizing from the solution in spatially distinct locations on the nucleation scaffold. This result is indicative of the complexity that develops in the FC system if DME exposure is extended to longer periods after the initial salt fraction has been crystallized. Moreover, the spatially distinct formation of dissimilar metal salt crystals lends to the hypothesis that solid product crystal structure is important to DME-FC separation factors.

18) Page 14, line 234- 236, it is stated that "These findings indicate that selective precipitations are influenced by initial solution composition, solute solubilities, process temperatures, and product crystal structure." This is a very general and unspecific conclusion and it is not at all clear how it relates to the results, it must be rephrased.

Thank you for your recommendation for improvement to this text. We have revised the passage to draw more specific conclusions:

Compositional data indicate that initial solution compositions affect separation factors of crystallizations. Moreover, solute solubilities are more deterministic of crystallized product than the solute concentration or its nearness to saturation. Process temperature also plays an important role; for example, at 20°C, transition metal sulfates are produced, while at 31°C, lanthanide sulfates are produced. The results also highlight the importance of product crystal structure, as it is possible to crystallize more than one metal salt within the same solid product (e.g., $\text{Fe}_{0.x}\text{Co}_{0.y}\text{SO}_4$).

19) Page 14, "fractional precipitation in stages" why not use "precipitation in stages" or "fractional precipitation" because fractional precipitation normally refers to precipitation in stages.

Thank you for the comment. This question is related to comment 8, which features a longer response. Here the operational definition of *fractional* describes the compositional difference between the solids produced and mother liquor as defined in the text.

20) On page 18 finally the presence of Fe(II) is mentioned. Why so late in the manuscript and what about oxidation to Fe(III)? You will have a mixture of Fe(II) and Fe(III)?

Thank you for bringing this to our attention. We anticipate iron is primarily present in these solutions in the +2 oxidation state, due to the magnet dissolution process, hydrolytic stability and the greater solubility of FeSO_4 compared to $\text{Fe}_2(\text{SO}_4)_3$. However, we did not quantify the solution for Fe^{2+} vs Fe^{3+} . We have changed language to remove the speculation with regard to the oxidation state of iron in solution.

We are currently conducting a study to examine the separation of Co and Fe with DME-FC by experimentation with both ferrous and ferric iron sulfates. This work, once complete, will shed light on the importance of oxidation state of iron in the separation of Fe and Co.

21) I like the approach in this manuscript and the idea of the paper but I think that the text must be improved and be given more thought. I am sorry for my bad phrasing due to lack of time. I hope that you will understand my comments.

Thank you for your time and for the insightful feedback.

Reviewer #2 (Remarks to the Author):

This manuscript deals with the selective recovery of critical metals from permanent magnet leachates by fractional precipitation. In general, precipitation methods in hydrometallurgy shows a poor selectivity in spite of simple operations. The solvent-driven fractional precipitation method in this manuscript has a metal selectivity; therefore, this is a promising method for environment-friendly metal refinings. Although the results are clearly presented, the authors should elaborate on the points listed below.

The authors would like to thank the reviewer for the analysis of the manuscript and recommendations for changes and future experimentation.

1) The solvent-driven fractional precipitation method can cause a reduction of the SX operation in the metal purification process. In contrast, the mutual separation of REE after the FP operation will still be conducted by SX. Hence, an efficient separation of Nd/Sm or Nd/Dy could make this method more eco-friendly. Does this method have a potential to achieve that?

Thank you for this recommendation; we agree that lanthanide/lanthanide separations are a worthy goal of DME-FC. It may be possible to achieve this separation directly with DME or via DME-FC or in combination with use of double salts. While use of a double salt would increase reagent consumption, double salts are often reusable and may improve Ln/Ln separation factors. We intend to pursue this line of research in the future.

A sentence in page 3 of the introduction reads:

Historically, REE separations were carried out through fractional crystallization, whereby evaporation or changes in solution temperature drive precipitation of individual REE double salts from the mixed REE salt solution.^{18,19}

18 Gupta, C. K. & Krishnamurthy, N. *Extractive Metallurgy of Rare Earths*. (CRC Press, 2005).

19 Enghag, P. *Encyclopedia of the Elements*. (Wiley-VCH, 2004).

2) I think that the metal concentration (component) in the obtained products (Fig 3) is significantly different from that which can be expected from Fig 2a. This is due to the contribution of DME molecules? DME can distinguish between transition metals and REEs?

Thank you for posing this question. Figure 2a presents temperature-dependent solubility limits in molal concentration for DME and the sulfates in solution, which we claim impacts the DME-FC process and allows us to crystallize *either* Sm-rich *or* Co-rich solids from the same initial leach solution in the case of the Sm-Co leachate. The exact composition of the product is also dependent on the initial compositions of the leachates (Figure 2b-c). In addition, the lattice plays a role in the purity of the crystallized products, as discussed in the Results and Discussion.

In response to Reviewer #2 question 4, we elaborate upon the role of DME in inducing crystallization:

Gaseous DME is employed as a saturation agent to induce **crystallization** of transition metal or lanthanide sulfates from mixed metal salt magnet leachates. **DME is not thought to interact directly with other solutes; instead, DME reduces the quantity of the free water (i.e., water that is not bound within a solvation environment) to fulfill its own hydration requirements.^{62,63} A reduction in free water, reduced water activity, and/or liquid phase microstructuring⁷⁹ induce salt precipitation. This mechanism suggests that the solid salt product compositions would be similar (if not identical) to those produced in energy-intensive evaporative precipitation processes conducted at equivalent temperatures.**

79 Marcus, Y. The structure of and interactions in binary acetonitrile + water mixtures. *Journal of Physical Organic Chemistry* **25**, 1072-1085, doi:10.1002/poc.3056 (2013).

3) The authors described that one mole of salt is replaced by one mole of MOS in their previous paper. Does this mole ratio depend on the metal salts?

The previous paper⁶⁰ presents that one mole of hydrated NaCl is displaced per mole of organic solute. We are certain it is a molar displacement phenomenon in the initial fractional crystallization of NaCl and there is strong likelihood that a similar trend exists in these transition metal and lanthanide sulfate systems near their binary saturation point, but one-to-one trend deviates as the ratio of DME:salt increases. This behavior was identified in

fractional crystallization of NaCl.⁶⁰ The binary CaSO₄:water saturation point is so dilute that the 1:1 displacement is unobservable.⁶⁰ These two results indicate that the observed phenomenon is governed by the solubility of the salt in solution. While not discussed in detail in this work, it is also likely that the hydration structure of DME and the individual salts also have an effect.

60 McNally, J. S. *et al.* Solute Displacement in the Aqueous Phase of Water-NaCl-Organic Ternary Mixtures Relevant to Solvent-Driven Water Treatment. *RSC Advances* **10**, 29516–29527 (2020).

4) The authors should explain a role of DME during the precipitation reaction in detail.

Thank you for requesting additional description of the role of DME in these fractional crystallization-based separations. We have elaborated upon the description of the mechanism of DME-driven FC on pages 6-7 at the start of the Results and Discussion section:

Gaseous DME is employed as a saturation agent to induce **crystallization** of transition metal or lanthanide sulfates from mixed metal salt magnet leachates. **DME is not thought to interact directly with other solutes; instead, DME reduces the quantity of the free water (i.e., water that is not bound within a solvation environment) to fulfill its own hydration requirements.^{62,63} A reduction in free water, reduced water activity, and/or liquid phase microstructuring⁷⁹ induce salt precipitation. This mechanism suggests that the solid salt product compositions would be similar (if not identical) to those produced in energy-intensive evaporative precipitation processes conducted at equivalent temperatures.**

79 Marcus, Y. The structure of and interactions in binary acetonitrile + water mixtures. *Journal of Physical Organic Chemistry* **25**, 1072-1085, doi:10.1002/poc.3056 (2013).

5) Why do the Nd-Fe-B leachates (Table S2) contain Sm? Nd-Fe-B magnets do not generally contain Sm.

Thank you for identifying this discrepancy. The Nd-Fe-B magnet leachate was produced from a mixed real-world industrial feed that contained some Sm-Co magnet to clarify the origin of Sm and Co in solution. The complexity of this leach solution is important, as it represents a real secondary feedstock where inputs have not been fully segregated. Magnet manufacturers process both magnet types and it is typical for generated secondary feedstock, e.g. swarf, to contain both Sm-Co and Nd-Fe-B magnets.

To clarify this point, we have made several modifications:

Changes to page 9 of the Results and Discussion section:

DME-FC was studied experimentally in conjunction with two separate REE-rich leachates: one leachate was produced from Sm-Co magnet swarf, whereas a more complex leachate was generated from a real-world mixed magnet recycling feed, containing both Nd-Fe-B and Sm-Co magnet swarf.

Changes to Page 19 of the Conclusions:

DME-driven fractional **crystallization was demonstrated** in the separation of rare earth element and transition metal salts from industrially generated magnet wastes. **DME-FC was applied to two separate** leachates, one comprising only Sm, Co, and Fe, and a more complex leachate containing Nd, Pr, Dy, Sm, Fe, and Co.

Throughout the figures and text, "Nd-Fe-B magnet leachate" has been replaced with "Nd-Fe-B **mixed** magnet leachate" to recognize the presence of Sm and Co in solution.

REVIEWERS' COMMENTS

Reviewer #1 (Remarks to the Author):

Thank you for your answers. According to me the manuscript can be published in its current state.

Reviewer #2 (Remarks to the Author):

The authors' explanations and modifications are acceptable.
I suggest that this revised manuscript be published in Nature Communications.